# Brief communication: How does complex terrain change the power curve of a wind turbine?

Niels Troldborg[1], Søren J. Andersen[1], Emily L. Hodgson[1], and Alexander Meyer Forsting[1]

[1]DTU Wind & Energy Systems, Frederiksborgvej 399, 4000, Roskilde, Denmark

**Correspondence:** Niels Troldborg (niet@dtu.dk)

**Abstract.** The power performance of a wind turbine in complex terrain is studied by means of Large Eddy Simulations (LES). The simulations show that the turbine performance is significantly different compared to what should be expected from the available wind. The reason for this deviation is that the undisturbed flow field behind the turbine is non-homogeneous and therefore results in a very different wake development and induction than seen for a turbine in flat homogeneous terrain.

## 1 Introduction

The power curve of a wind turbine shows the relationship between its power output and the undisturbed wind speed at hub height. In combination with estimates of the wind resource, the power curve is used to predict the expected energy yield of a wind turbine at a candidate site. Thus, the power curve is one of the most important characteristics of a wind turbine and therefore is also typically warranted by the manufacturer. Power performance verification tests are usually conducted in flat homogeneous terrain where the undisturbed wind speed is approximated by measuring sufficiently far upstream (typically 2.5 rotor diameters). In complex terrain this approach is invalid because the upstream flow in this case is not homogeneous. Instead it is common practice to perform a site calibration prior to erecting the turbine in which the wind speed at the location of the turbine is related to the corresponding wind speed measured at some upstream location.

An alternative to site calibration is to use a nacelle mounted lidar to measure at several ranges closer to the rotor and make proper corrections of the measured flow to account for the induction effect (Borraccino et al., 2017).

In either case the idea is to establish the free-stream conditions at the position of the turbine. Most work on wind turbine power performance verification in complex terrain focus on how to establish a robust and accurate free-wind speed estimate and thereby reduce the scatter in the power curve (Brodeur and Masson, 2008; Nam et al., 2004; Borraccino et al., 2017). However, to the best of our knowledge, Oh and Kim (2015) are so far the only ones to investigate whether the power curve of a turbine is identical in flat and complex terrain. They analyzed the actual measured power curve of five wind turbines in a wind farm at a complex site and found large differences between the turbines as well as with the power curve guaranteed by the manufacturer. They used the different power curves to predict the annual energy production (AEP) and found that the estimates based on the measured curves could be up to 17.8% lower than when using the guaranteed power curve.

A disadvantage of using field measurements to analyze power performance in complex terrain is that there inevitably will be uncertainties in the predicted power curve. The biggest uncertainty lies in determining the free-stream velocity but a turbine

may also perform differently than expected due to e.g. erosion, icing or blade surface contamination. In addition the stochastic nature of the wind resource requires very long measurement periods to get converged statistics.

Simulations on the other hand do not have these issues and therefore are ideal for studying power curves and how they may change in complex terrain. Furthermore, it has been shown that simulations using both RANS (Allen et al., 2020; Sessarego et al., 2018) and LES (Liu et al., 2021; Yang et al., 2018; Shamsoddin and Porté-Agel, 2017) are reasonably accurate at predicting the power performance of wind turbines in complex terrain.

However, the question as to how the terrain impacts the power curve of a wind turbine still remains unanswered. The objective of the present work is to answer this question by conducting LES of the power performance of a wind turbine in complex terrain and comparing with the corresponding predictions in flat terrain.

## 2  Methodology

In the following we consider a DTU-10MW wind turbine (Bak, 2013) operating in both flat and complex terrain. This turbine has a diameter of 178.34 m and a hub height of 119 m. The complex terrain is based on the topography at the site of Perdigão in Portugal consisting on two parallel ridges and the turbine is in this case placed on top of the first ridge. The ratio between ridge height and turbine diameter is 1.5. The curvilinear grid used to resolve the terrain is described in Berg et al. (2017) except that here it is extended with a flat region after the terrain where the turbulence is allowed to dampen out before exiting the domain. The dimensions of both computational domains are $L_x \times L_y \times L_z = 8480$ m$\times$2560 m$\times$3081 m, where subscripts $x$, $y$ and $z$ refer to the streamwise, spanwise and vertical directions, respectively. In both cases the number of grid cells in each direction is $512 \times 256 \times 256$. In the first part of the domains ($x \leq 4640$ m) and close to the surface, the grid cells have dimensions $dx = dy = 2dz = 10$ m. The cells are gently stretched in the vertical direction and for $x > 4640$ m, they are also stretched towards the outlet boundary.

The inlet to the simulations is determined in a separate precursor simulation where the flow is driven over a flat rough surface by a constant pressure gradient and the flow is assumed fully neutral. The precursor grid is 5120 m long and has a streamwise grid spacing of $dx = 10$ m, while its cross-section is identical to the inlet boundary of the main grids. In the precursor the friction velocity is 0.3 m/s and the roughness height is $z_0 = 2 \cdot 10^{-4}$ m. However, a wide range of different inflow conditions are generated by transforming the data from the simulation as follows:

$$u^{new} = u_*^{new} \left( \frac{u^{org}}{u_*^{org}} + \frac{1}{\kappa} \ln \frac{z_0^{org}}{z_0^{new}} \right) \tag{1}$$

where superscript "org" refers to the original precursor field. The above transformation is valid for rough-wall boundary layers at high Reynolds numbers in which the roughness elements are much smaller than the boundary-layer height (Castro, 2007).

The wind turbine is modelled as an actuator disk (AD) combined with the aero-elastic model Flex5 (Øye, 1996). All simulations are carried out as LES using the incompressible Navier-Stokes solver EllipSys3D (Sørensen, 1995) and the sub-grid stresses are modelled using the closure by Deardorff (1980).

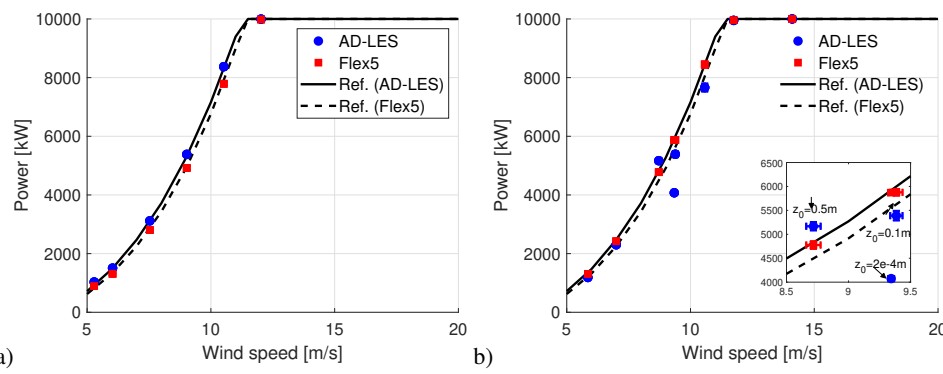

**Figure 1.** Power curve of DTU-10MW turbine in flat (a) and complex (b) terrain as predicted by AD-LES and Flex5, respectively. Note that the error bars indicating the standard error of the mean are included but they are barely visible.

## 3 Results

In the following we present results from a series of simulations where $u_*$ is varied between $0.2$ m/s and $0.6$ m/s and the roughness height is varied between $2 \cdot 10^{-4}$ m and $0.5$ m. More details about the wind turbine inflow characteristics for each
case are provided in the appendix. In each case we simulate 1.5 hours of real time flow but only analyse the last hour in order to get rid of any initial transients. Each 1 hour simulation is split into $6 \times 10$ minute sections from which we compute ensemble averaged 10 minute statistics and evaluate the variability via the standard error of the mean.

Fig. 1 shows the power curve of the turbine in complex and flat terrain as predicted by AD-LES and standalone Flex5, respectively (markers). The inflow for the standalone Flex5 simulations are extracted from the LES cases without the turbine
included. As reference (black lines), the power curves predicted by both methods at uniform laminar inflow are also included. Note that the AD-LES reference curve is computed on a cubic grid as described by Hodgson et al. (2021), but with a grid resolution which is similar to the one used here for resolving the terrain.

The power predicted by AD-LES in flat terrain (below rated wind speed) is about 10% higher than what is found using standalone Flex5, but in both cases the predictions are in good agreement with their respective reference power curves. The
difference between AD-LES and Flex5 is primarily due to the rather coarse grid resolution used here (Hodgson et al., 2021).

In the complex terrain case, the power predicted by AD-LES differ significantly from both the reference power curve and the Flex5 predictions. In most cases the AD-LES predicts a power output which is 10%-15% below the reference power curve, but in one case it is more than 30% below and in another case the power is above the reference power curve. This behaviour can be explained by the non-homogeneous development of the free-stream flow field behind the turbine and how it is affected
by surface roughness: A deceleration of the free-stream flow behind the turbine will cause a slower transport velocity of the wake and therefore a larger induction in the rotor plane, which in effect will reduce the power output of the turbine compared to the flat terrain counterpart. Conversely, an acceleration of the free-stream flow field behind the turbine should augment the

expected power output[1]. This mechanism is not captured by the Flex5 simulations because it inherently assumes the turbine to operate in a homogeneous flow.

To verify this explanation, Fig. 2 shows contours of the mean streamwise velocity with and without the turbine included for the three cases at wind speeds between $8.5$ m/s and $9.5$ m/s, which are highlighted in Fig. 1. These cases mainly differ by their surface roughness, which in effect causes a very different free-stream flow field behind the turbine as seen in Fig. 2a)-c).

At $z_0 = 0.5$ m there is a large separated region behind the ridge, which acts as a barrier and therefore pushes the flow passing over the hill upwards. As a consequence the free-stream velocity initially accelerates downstream of the rotor and, as shown in Fig. 2f), this causes a weaker wake leading to lower induction in the rotor plane. Consequently the power increases above the reference power as expected.

As the roughness is decreased the separated region behind the ridge becomes smaller and smaller and eventually the flow becomes nearly attached to the terrain surface. In the two lower roughness cases the flow therefore decelerates immediately downstream of the turbine and as seen in Fig. 2d)-e) this leads to stronger wakes and hence the power output reduces compared to the flat terrain counterpart.

The strong impact that the flow development in the lee of the ridge has on the wake and induction is consistent with the findings by Meyer Forsting et al. (2016).

In order to get a more quantitative impression of the mechanisms described above, Fig. 3 shows the free-stream velocity ($U_0$) and induction ($U_0 - U$) along the centreline of the turbine for the three cases shown in Fig. 2. The figure clearly shows that the induction in the rotor plane correlates with the level of acceleration/deceleration of the free-stream velocity downstream of the turbine: a strong deceleration leads to strong induction and vice versa.

## 4  Discussion

The results presented above show us that the power curve of a turbine in complex and flat terrain may differ significantly from each other. Although it may seem surprising at a first glance it is qualitatively in good agreement with the work of Oh and Kim (2015). In addition, there is no contradiction between this finding and some of the theories on diffuser augmented rotors. For example Jamieson (2009) showed that the theoretical maximum power coefficient of a turbine in a diffuser is

$$C_{p,max} = \frac{16}{27}(1 - a_0) \tag{2}$$

where the $16/27$ is recognized as the Betz limit and $a_0$ is the induction parameter due to the diffuser at the position of the rotor. Since the diffuser produces a speed-up ($a_0 < 0$) Eq. 2 predicts an augmented performance of the turbine. However, in Eq. 2 the power coefficient is based on the undisturbed velocity far upstream, $U_0$. To express the performance analogously to the complex terrain case, we need to base the power coefficient on the free-stream velocity at the position of the rotor, i.e.

---

[1]Alterations in transport velocity have also previously been identified to change the rotor induction in wind farms and complex terrain (Meyer Forsting et al., 2016; Meyer Forsting et al., 2017).

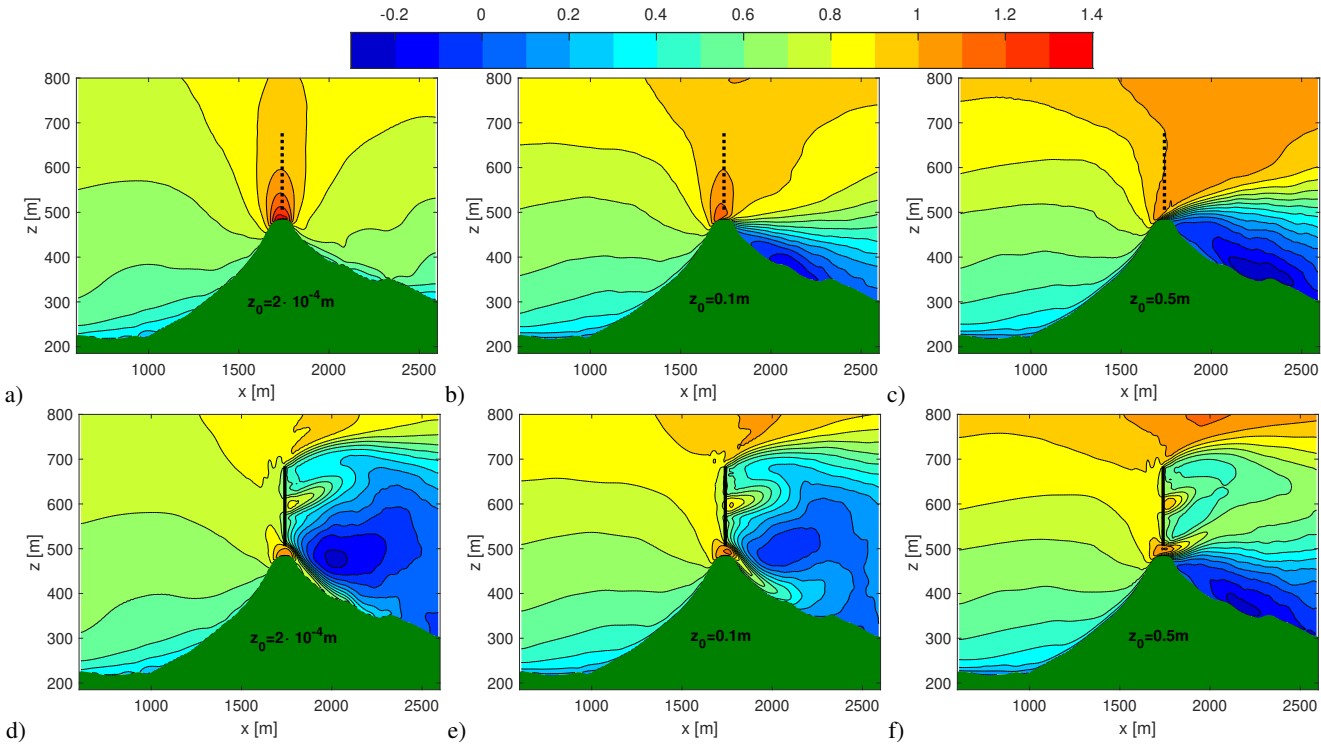

**Figure 2.** Contours of the mean streamwise velocity without (top) and with (bottom) the turbine included at different surface roughness levels. The velocities are scaled with the free-stream velocity at the hub position of the wind turbine.

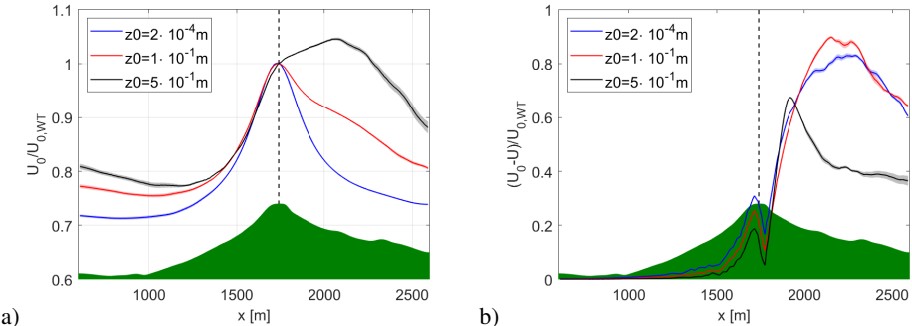

**Figure 3.** Free-stream velocity (a) and induced velocity (b) along the centerline of the turbine at different roughness heights. The velocities are scaled with the free-stream velocity at the position of the wind turbine. The shaded area indicate the standard deviation of the mean. The vertical dashed line indicates the position of the turbine

$U_0(1 - a_0)$. In that case the maximum achievable power coefficient becomes

$$C_{p,max} = \frac{\frac{16}{27}}{(1 - a_0)^2} \tag{3}$$

which is lower than the Betz limit when $a_0 < 0$.

Besides the topography itself, our work also shows that the power performance is strongly governed by the roughness of the terrain. Although not investigated in the present work, we expect that atmospheric stability will also have a very strong impact on the performance of the turbine because it affects the level of separation behind the hill and hence also the extent to which the wake follows the terrain as shown by Menke et al. (2018).

The consequence of the above findings is that a site calibration may not be sufficient when verifying the power performance of turbines in complex terrain. Even in cases where the bias shown here in practice will average out during a full site calibration campaign (due to variations in atmospheric stability, wind conditions and seasonal changes in roughness) it is clear that disregarding of the downstream development will lead to increased uncertainties in the power curve verification. In general the power curve of a turbine is site specific and hence in principle a performance verification should be carried out for each individual site or at least a proper correction should be adopted. This does not only pertain to turbines in complex terrain but will apply whenever the ambient flow is non-homogeneous including in wind farms as also shown by Meyer Forsting et al. (2017).

## 5   Conclusions

The power performance of a DTU-10MW turbine located in a complex terrain has been studied via large eddy simulations. The simulations revealed that the power curve for the turbine was significantly different than for the same turbine in flat homogeneous terrain. The reason for this difference is that the undisturbed velocity in the region behind the turbine becomes non-homogeneous at the complex site and therefore the wake deviates significantly from that generated when the turbine is operating in flat terrain. Thus, the answer to the question posed in the title is that if the terrain causes an deceleration of the free-stream flow behind the turbine then it leads to under performance of the turbine, whereas the opposite is true for a downstream flow acceleration. The magnitude of the power curve modification depends on how much the free-stream flow varies behind the turbine, which again depends on both the roughness and terrain topography. As a consequence the power curve cannot be seen as a unique characteristic of a turbine but will be site specific.

## Appendix A:  Characteristics of wind turbine inflow

Tables A1-A2 show some characteristics of the inflow seen by the wind turbine for each case. The entities in the tables are: the friction velocity $u_*$, roughness height $z_0$, hub velocity $U_{hub}$, turbulence intensity $TI$, vertical inflow angle $\theta$, shear exponent $\alpha$ and veer $\phi$. Both $\alpha$ and $\phi$ are computed from the velocities at lower and upper tip height.

As seen there is a mild sensitivity of the results to friction velocity for a given roughness. This is unexpected because the

flow should be Reynolds independent. However, it can be explained from 1) limited effective grid resolution, which affects the sub-grid scale turbulence level and 2) statistical sensitivity, which stems from the fact that the averaging time is the same in all cases and therefore the number of flow-through times varies with friction velocity.

**Table A1.** Inflow characteristics for each case in complex terrain.

| Case | $u_*[m/s]$ | $z_0[m]$ | $U_{hub}$ | $TI[\%]$ | $\theta[^o]$ | $\alpha$ | $\phi[^o]$ |
|------|-----------|----------|-----------|----------|--------------|----------|------------|
| 1 | 0.2 | $2 \cdot 10^{-4}$ | 9.3 | 2.4 | 1.8 | -0.14 | 0.2 |
| 2 | 0.25 | $1 \cdot 10^{-1}$ | 5.9 | 4.8 | 5.8 | -0.079 | 2.9 |
| 3 | 0.3 | $1 \cdot 10^{-1}$ | 7.0 | 4.6 | 6.1 | -0.076 | 2.5 |
| 4 | 0.4 | $1 \cdot 10^{-1}$ | 9.4 | 5.2 | 5.8 | -0.076 | 2.6 |
| 5 | 0.45 | $1 \cdot 10^{-1}$ | 10.6 | 5.3 | 5.6 | -0.080 | 2.7 |
| 6 | 0.5 | $1 \cdot 10^{-1}$ | 11.7 | 5.5 | 5.6 | -0.079 | 2.8 |
| 7 | 0.6 | $1 \cdot 10^{-1}$ | 14.1 | 5.9 | 5.4 | -0.079 | 2.6 |
| 8 | 0.5 | $5 \cdot 10^{-1}$ | 8.7 | 7.0 | 11.0 | -0.015 | 6.9 |

**Table A2.** Inflow characteristics for each case in flat terrain.

| Case | $u_*[m/s]$ | $z_0[m]$ | $U_{hub}$ | $TI[\%]$ | $\theta[^o]$ | $\alpha$ | $\phi[^o]$ |
|------|-----------|----------|-----------|----------|--------------|----------|------------|
| 1 | 0.2 | $2 \cdot 10^{-4}$ | 6.0 | 4.8 | -0.11 | 0.11 | 0.22 |
| 2 | 0.25 | $2 \cdot 10^{-4}$ | 7.5 | 5.1 | -0.00 | 0.10 | 0.31 |
| 3 | 0.3 | $2 \cdot 10^{-4}$ | 9.0 | 5.2 | -0.03 | 0.10 | 0.47 |
| 4 | 0.35 | $2 \cdot 10^{-4}$ | 10.5 | 5.5 | -0.01 | 0.097 | 0.45 |
| 5 | 0.4 | $2 \cdot 10^{-4}$ | 12.0 | 5.6 | -0.01 | 0.097 | 0.40 |
| 6 | 0.5 | $5 \cdot 10^{-1}$ | 5.3 | 13.0 | -0.01 | 0.18 | 1.6 |

*Author contributions.* NT performed the simulations and drafted the article. SJA and AMF contributed to the idea and methodology. ELH and SJA performed code validation. SJA, ELH and AMF supported the analysis, review and edited the manuscript

*Competing interests.* The authors declare that they have no conflict of interest.

*Acknowledgements.* The Danish Energy Agency is acknowledged for their support via the IEA Wind Task 32 project, Journal No. 64019-0519.

We would like to thank Hans Ejsing Jørgensen for leading the project.

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
