# Peer review of "Brief communication: How does complex terrain change the power curve of a wind turbine?"

_Wind Energy Science, 2022_

## Author Response (AR1)

Dear Editor,

Thank you very much for arranging the review of our manuscript. We also want to express our sincere gratitude to the reviewers for reviewing our manuscript. They were very helpful in pointing out the weak and unclear parts of the manuscript. Based on the reviews, we have made several modifications of the original manuscript and thereby improved it significantly. The biggest change is the addition of an appendix with tables presenting some important characteristics of the wind turbine inflow.

We hope that you will find the new revision ready for publication in WES.

Please find below our response (in blue) to the reviewers comments (in black).

**Reviewer 1**

Interesting case study discussing the differences between power curves in flat vs complex terrain supporting the need for site-specific calibration. The numerical approach and results are well described and solid. The only limitation I see is on the statistical significance of the assessment, with only 6 10-min samples per case to compute ensemble averages. Also, the generalization of the simulation results to real world campaigns needs to be discussed a bit more. In particular, the limited representativeness of the simulations when we compare with the statistis of a site calibration campaign that averages a much wider range of flow cases. Nevertheless, I would agree with the general conclusion of the study, that power curves are site specific, with good examples provided in this study. I believe this is a well-established conclusion in the wind industry. Please elaborate more on the statistical aspects to enrich the discussion of the case study.

Comments:

P2.35: Please specify the hub-height and rotor diameter of the reference turbine

This is now specified by adding the following sentence in the beginning of section 2:

*This turbine has a diameter of 178.34 m and a hub height of 119 m.*

P2.42: "the grid cells has dimensions" ¿ the grid cells have dimensions

Noted and changed

P3.58: In addition to the roughness length and friction velocity it would be useful to present the hub-height wind speed, turbulence intensity, inflow angle and rotor-based wind shear (power-law exponent) and wind veer. These are more useful to understand the range of conditions in terms of siting parameters. Can you provide a table with this info?

Good point!. We have now added the following two tables (one for each of the considered terrains) with these info in Appendix in order not to exceed the number of allowed tables/figures in a brief communication.

We refer to the tables in the beginning of section 3 via the following sentence:

*More details about the wind turbine inflow characteristics for each case are provided in the appendix.*

In the appendix the tables are accompanied with the following text, which both explains the used nomenclature and why the results in the tables show a mild but unexpected sensitivity to friction velocity for a given roughness:

*Tables 1-2 show some characteristics of the inflow seen by the wind turbine for each case. The entities in the tables are: the friction velocity $u_*$, roughness height $z_0$, hub velocity $U_hub$, turbulence intensity $TI$, vertical inflow angle $\theta$, shear exponent $\alpha$ and veer $\phi$. Both $\alpha$ and $\phi$ are computed from the velocities at lower and upper tip height.*

*As seen there is a mild sensitivity of the results to friction velocity for a given roughness. This is unexpected because the flow should be Reynolds independent. However, it can be explained*

Table 1: Inflow characteristics for each case in complex terrain.

| Case | $u_*[m/s]$ | $z_0[m]$ | $U_{hub}$ | $TI[\%]$ | $\theta[^o]$ | $\alpha$ | $\phi[^o]$ |
|------|------------|----------|-----------|----------|--------------|----------|------------|
| 1 | 0.2 | $2 \cdot 10^{-4}$ | 9.3 | 2.4 | 1.8 | -0.14 | 0.2 |
| 2 | 0.25 | $1 \cdot 10^{-1}$ | 5.9 | 4.8 | 5.8 | -0.079 | 2.9 |
| 3 | 0.3 | $1 \cdot 10^{-1}$ | 7.0 | 4.6 | 6.1 | -0.076 | 2.5 |
| 4 | 0.4 | $1 \cdot 10^{-1}$ | 9.4 | 5.2 | 5.8 | -0.076 | 2.6 |
| 5 | 0.45 | $1 \cdot 10^{-1}$ | 10.6 | 5.3 | 5.6 | -0.080 | 2.7 |
| 6 | 0.5 | $1 \cdot 10^{-1}$ | 11.7 | 5.5 | 5.6 | -0.079 | 2.8 |
| 7 | 0.6 | $1 \cdot 10^{-1}$ | 14.1 | 5.9 | 5.4 | -0.079 | 2.6 |
| 8 | 0.5 | $5 \cdot 10^{-1}$ | 8.7 | 7.0 | 11.0 | -0.015 | 6.9 |

Table 2: Inflow characteristics for each case in flat terrain.

| Case | $u_*[m/s]$ | $z_0[m]$ | $U_{hub}$ | $TI[\%]$ | $\theta[^o]$ | $\alpha$ | $\phi[^o]$ |
|------|------------|----------|-----------|----------|--------------|----------|------------|
| 1 | 0.2 | $2 \cdot 10^{-4}$ | 6.0 | 4.8 | -0.11 | 0.11 | 0.22 |
| 2 | 0.25 | $2 \cdot 10^{-4}$ | 7.5 | 5.1 | -0.00 | 0.10 | 0.31 |
| 3 | 0.3 | $2 \cdot 10^{-4}$ | 9.0 | 5.2 | -0.03 | 0.10 | 0.47 |
| 4 | 0.35 | $2 \cdot 10^{-4}$ | 10.5 | 5.5 | -0.01 | 0.097 | 0.45 |
| 5 | 0.4 | $2 \cdot 10^{-4}$ | 12.0 | 5.6 | -0.01 | 0.097 | 0.40 |
| 6 | 0.5 | $5 \cdot 10^{-1}$ | 5.3 | 13.0 | -0.01 | 0.18 | 1.6 |

*from 1) limited effective grid resolution, which affects the sub-grid scale turbulence level and 2) statistical sensitivity, which stems from the fact that the averaging time is the same in all cases and therefore the number of flow-through times varies with friction velocity.*

P3.72: With only 6x10-min samples, have you reached converged statistics to make an unbiased assessment of the differences between flat and complex terrain? Please justify if 1.5 hr simulation time is long enough.

The variability in the power estimates is small so we are confident that our simulations are long enough to get an unbiased assessment. We agree with you that 6 samples of 10 minutes may not be enough to establish the fully converged statistical difference between AD-LES and Flex-standalone. However, our aim here is only to show that there can be a significant terrain induced difference between the two types of predictions and to give an impression of how big this difference can be.

It should be emphasized that the inflow used for the AD-LES and Flex-standalone is exactly the same for each 1.5hr case and thereby we overcome many of the potential issue related to lack of statistical convergence. When comparing the individual 10 minute predictions we of course find some variability but still the offset between AD-LES and Flex-standalone is very consistent. This is illustrated in the plots below, which shows the 6x10-min samples for three selected cases at different roughness levels. In order to show that our analysis is not affected by lack of statistical convergence, we have now added error bars in the power curve plots. In addition we have added the following wording in the caption to Fig. 1:

*Note that the error bars indicating the standard error of the mean are included but they are barely visible.*

P4.Fig2: The velocities are scaled with the free-stream velocity at the position of the wind

[Figure]

Figure 1: Plot of 10 minute averaged power as a function 10 minute averaged free-stream velocity. From left to right: a) $z_0 = 2 \cdot 10^{-4}m$, b) $z_0 = 1 \cdot 10^{-1}m$ and c) $z_0 = 5 \cdot 10^{-1}m$

turbine ¿ at hub-height right? I would add a third row with the difference between the two contour plots, without and with turbine, to highlight and quantify differences more clearly.
Yes, the contours are scaled with the velocity at the hub position of the rotor. We have clarified this by changing the last part of the caption in Fig. 2 from:
*The velocities are scaled with the free-stream velocity at the position of the wind turbine.*
to
*The velocities are scaled with the free-stream velocity at the hub position of the wind turbine.*
Thanks for the advice of adding an additional row of plots. These plots are shown in the figure below. As seen these plots essentially show the same flow behaviour as Fig. 2 d-f and therefore

[Figure]

Figure 2: Contours of the mean streamwise induced velocity for different surface roughness levels. The induced velocities are scaled with the free-stream velocity at the hub position of the wind turbine.

are not adding much more value to understanding the flow mechanisms at play. Therefore, we have decided not to do include them in the article.

P4.82: "and, as shown in Fig. 2f), this causes" (add commas)
Noted and changed

P4.92: For completeness, can you add the formula for induced velocity that you use to plot Fig.3b?
We have now added the formula in the text when presenting Fig. 3:
*... Fig. 3 shows the free-stream velocity $(U_0)$ and induction $(U_0 - U)$ along the centreline of the turbine ....*
The formula is also shown in the figure label.

P5.111: "... a site calibration cannot stand alone when verifying the power performance of turbines in complex terrain". I wouldn't reach this conclusion without a more complete assessment that would replicate the statistics that you gather in a IEC 61400-12 site calibration campaign. This study shows that the differences can be large in a few 10 min samples but these

differences may average out when you use a large number of samples over a period that captures a representative range of stability, wind speed and direction changes.

We agree with you that in real life the differences may average out due to variations in stability, wind speed, wind direction, roughness etc. However, our work shows that it is problematic to use a site calibration because it only accounts for the inhomogeneity in the upstream flow and not in the downstream flow. By not accounting for the downstream development there will be a bias (which depends on the atmospheric flow and terrain) and even if this bias in practice average out, it will still lead to increased uncertainties in the power curve verification.

In order to meet your comment We have rephrased the sentence to:

*The consequence of the above findings is that a site calibration may not be sufficient when verifying the power performance of turbines in complex terrain. Even in cases where the bias shown here in practice will average out during a full site calibration campaign (due to variations in atmospheric stability, wind conditions and seasonal changes in roughness) it is clear that disregarding the downstream development will lead to increased uncertainties in the power curve verification.*

P6.114: including in wind farms ¿ I would rather say "including in waked conditions"

The effect shown in our article will affect all wind turbines in a wind farm and not only those who are waked. This is e.g. shown in the article by Meyer Forsting et al. "The flow upstream of a row of aligned wind turbine rotors and its effect on power production".

Therefore, we will stick to the formulation "...including in wind farms"

**Reviewer 2**

The paper is interesting, relevant and well-written. It is usually accepted that in reality the power curves differ from theoretical, but it is nice to see this question addressed in literature. The brevity of the paper is commendable.

General comments:

I do not understand the number and configuration of numerical experiments carried out. P3L56 states: "In the following we present results from a series of simulations where u is varied between 0.2 m/s and 0.6 m/s and the roughness height is varied between 2·104 m and 0.5 m." I assume that different friction length values give different wind speeds for Figure 1. The inset of Figure 1 shows points with different roughness length values. But what roughness length values were used for other points in the plot? Were different roughness lengths investigated for wind speeds between 10 and 12 m/s?

This has now been clarified by adding tables with the inflow characteristics to the turbine for each simulated case. These tables are shown in appendix. See also our answer to reviewer 1's third comment.

P3L59: "From the simulation we compute ensemble averaged 10 minute statistics and evaluate the variability via the standard error of the mean". What is meant by "ensemble" here? 10 minute time-averaged values? Ensemble over different roughness lengths? It seems not, according to Figure 1. Also, I do not see any indications of variability in the Figure 1. Is the error-bars too small to be seen?

What is meant is that we split each 1 hour simulation in 6x10 minute sections and compute the 10 min statistics. This is now clarified by rephrasing the sentence to:

*Each 1 hour simulation is split into $6 \times 10$ minute sections from which we compute ensemble averaged 10 minute statistics and evaluate the variability via the standard error of the mean.*

The error bars are now included in Fig. 1 but they are barely visible.

P4L81: "At z0 = 0.5 m there is a large separated region behind the ridge, which acts as a barrier and therefore pushes the flow passing over the hill upwards. As a consequence the free-stream velocity initially accelerates downstream of the rotor and as shown in Fig. 2f) this causes a weaker wake leading to lower induction in the rotor plane". I suggest rewriting these sentences. I think I agree with the authors, but the phrasing is confusing. My problem is with the fact that "flow seperation" traditionally means seperation from the surface, clearly seen in (b), (c) and (f). But can we really talk about flow separation if the flow is following the surface in (d) and (e)? I would suggest calling the low wind speed region in (d) and (e) "wake region". I am not sure if the wake region is smaller in (d) than in (f). I am also wondering if the existence of flow along the associated with under the curve performance.

What we write is that there is a large separated region behind the ridge when $z_0 = 0.5$ $m$ and that this separation bubble shrinks when the roughness decrease. You are right that at the lowest roughness the free flow is nearly attached so here there is no apparent flow separation. To clarify this we have rephrased the sentence about the shrinking separation region to:
*As the roughness is decreased the separated region behind the ridge becomes smaller and smaller and eventually the flow becomes nearly attached to the terrain surface.*
In addition we have changed separation to flow development in L90 P4.
The wind turbine wake is smaller in f) than in d) because of the accelerating background flow just behind the turbine in the former case.
The question as to whether the potential flow along the ridge (if we understand correctly) will affect the rotor performance is valid but remember that the inflow to the Flex-standalone simulation is exactly the same as that of the AD-LES so if this was the case then we should see it in both types of simulations.

Minor comments:
I would suggest adding hub-height and turbine extent to the Figure 2.
The turbine position is indicated as dashed/full lines in Fig. 2. The hub height and diameter is now also specified. See our answer to the first comment from reviewer 1.

P2L31: "However, the question as to how the terrain impacts the power curve of a wind turbine still remains unanswered. The objective of the present work is to answer this question." I am afraid that authors can only partially answer this question as large number of terrain configurations remain not investigated.
We completely agree with the reviewer that we are not fully answering this question. However, we do show how the performance is affected by the non-homogeneous free-flow behind the turbine, i.e. acceleration leads to over performance and deceleration to under performance. This is now clarified by adding the following wording at the end of the conclusion:
Thus, the answer to the question posed in the title is that if the terrain causes an deceleration of the free-stream flow behind the turbine then it leads to under performance of the turbine, whereas the opposite is true for a downstream flow acceleration. The magnitude of the power curve modification depends on how much the free-stream flow varies behind the turbine, which again depends on both the roughness and terrain topography. As a consequence the power curve cannot be seen as a unique characteristic of a turbine but will be site specific.

**Additional changes**

P4: Added a footnote with reference to previous work which also shows how alterations in transport velocity change the rotor induction in wind farms and complex terrain
P2, L26: Added the following wording:

*In addition the stochastic nature of the wind resource requires very long measurement periods to get converged statistics.*
to emphasize another challenge of studying power curves from field measurements.